# [1,n]-Metal migrations for directional translational motion at the molecular level

Emma L. Hollis[1], Michael N. Chronias[1], Carlijn L. F. van Beek [1,2], Paul J. Gates[1] & Beatrice S. L. Collins [1] ✉

The controlled translational motion displayed by nature's motor proteins underpins a wealth of processes integral to life, from organelle transport to muscle contraction. The motor proteins move along one dimensional cytoskeletal tracks, with their motion characterised by high association of the enzyme to the biopolymer combined with highly dynamic motion along the track. Here we introduce carbon-to-carbon metal migration as a platform for dynamic association and show how such migrations, in combination with the incorporation of a simple hydrocarbon, can be harnessed to achieve autonomous directional translational motion of a metal centre along the length of a polyaromatic track.

Efforts to mimic the dynamic association displayed by nature's motor proteins and control translational motion in fully synthetic systems have been dominated by rotaxane-based systems[1-4], where the properties of the mechanical bond ensure complete association between the moving component (the macrocycle) and the track it encircles, while allowing high rates of translation through shuttling of the moving component under Brownian motion[5]. In addition to the dynamic association displayed by many rotaxane systems, by careful design of the track and macrocyclic component, elegant strategies have been employed to further control the motion in these mechanically interlocked systems, with both energy and information ratchet mechanisms allowing directional translational motion to be achieved[6-16]. Other than mechanical bonds, alternative platforms for achieving controlled translational motion in fully synthetic systems have had more limited success[17], with bipedal walker systems that exhibit dynamic association lacking mechanisms to achieve inherent directionality[18-20], and bipedal systems that do display high levels of directionality requiring stepwise intervention of an experimentalist (i.e., they lack the dynamic autonomous behaviour that underpins nature's walkers)[21-24].

Carbon-to-carbon metal migrations have been reported extensively in the literature over the last 25 years and have become established as a powerful method for remote functionalisation and the construction of complex polycarbocyclic structures[25-28]. Palladium and rhodium migrations have been most extensively developed[29-40], although other transition metals have also been shown to undergo migration[41-45], and migrations between carbon atoms four or five

carbon atoms apart, termed [1,4] and [1,5] migrations, respectively, dominate the field due to the energetically accessible five- and six-membered metallocycles through which these pathways pass. Extensive mechanistic studies suggest that for palladium and rhodium, the metal centre remains covalently bound to the substrate through the migration[29-40], giving rise to highly reversible processes with high rates of reaction, characteristics which lead us to propose that metal migrations could be exploited to achieve the dynamic association required for mimics of nature's motor proteins. In addition to exploiting the dynamic association exhibited by metal migrations, we use the incorporation of a simple hydrocarbon to ratchet the system and achieve the directional translation of a rhodium centre over extended polyaromatic tracks.

On examination of the archetypal polyaromatic track, 1,4-polyphenylene, we reasoned that, in its simplest form, directional translational motion of a metal centre along the track would comprise alternating [1,2] and [1,4] C(sp²)-to-C(sp²) metal migrations, with the metal centre moving from site A to site B and site B to site A′, and so on (Fig. 1a; once the symmetry of the track in broken, i.e., if the metal is associated with the track and does indeed move directionally along it, sites A and B are no longer degenerate). Such behaviour as the basis of directional motion along the polyaromatic chain, however, is precluded in two ways. Firstly, while [1,4] and [1,5] metal migrations are described extensively in the literature, the corresponding [1,2] metal migrations are scarcely reported, with a single example of a [1,2] rhodium migration having been described[40]. Secondly, the aryl-to-aryl

[1]School of Chemistry, University of Bristol, Cantock's Close, Bristol BS8 1TS, UK. [2]Stratingh Institute for Chemistry, University of Groningen, Groningen 9747 AG, The Netherlands. ✉e-mail: bs.lefanucollins@bristol.ac.uk

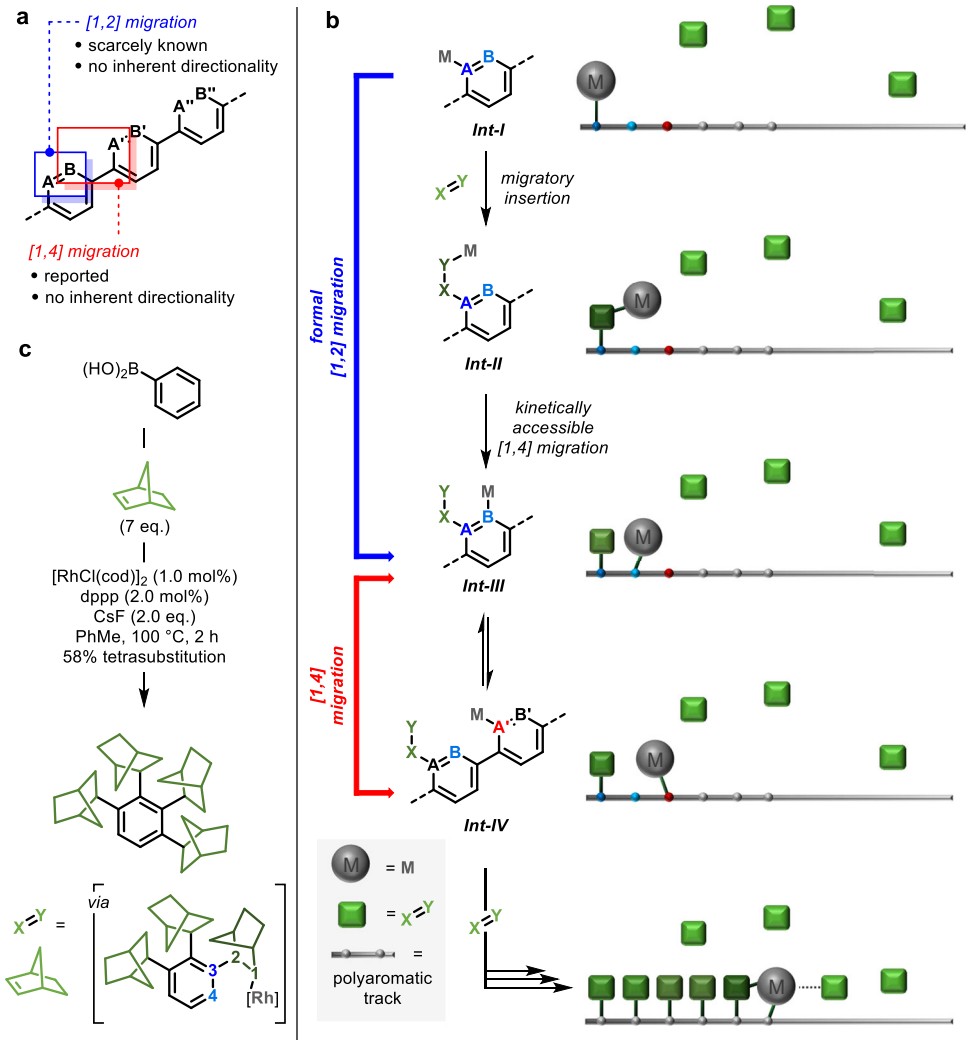

**Fig. 1 | Design for controlled translational motion. a** Alternating [1,2] and [1,4] C(sp²)-to-C(sp²) metal migrations for translational motion of a metal centre along a 1,4-polyphenylene track. **b** Design for controlled translational motion of a metal centre through repeating sequence of migratory insertion–[1,4] migration–[1,4] migration. **c** Miura's Rh(I)-catalysed iterative C–H alkylation with norbornene[35];

note that a single diastereoisomer of the tetra-substituted product is depicted with incorporated norbornane units depicted arbitrarily as (R,R,S). Red and blue used to highlight [1,2] and [1,4] C(sp²)-to-C(sp²) metal migrations; green used to illustrate sequential incorporation of norbornene.

migrations described in Fig. 1a lack inherent directionality: the species before and after the migrations, each with one C(sp²)–metal bond and one C(sp²)–H bond, have little to differentiate them (the structures with the metal at site A versus B, and B versus A′) and thus, while the metal centre may indeed move with dynamic association along the length of the polyaromatic track, it would do so bidirectionally with no inherent directionality.

To address these two issues and achieve directional motion of the metal centre along the polyaromatic track, we recognised that the incorporation of a simple hydrocarbon would allow us to achieve 1) a general solution to the kinetically inaccessible migration of a metal centre between two adjacent carbon atoms, and 2) a symmetry breaking of the migrations such that they become biased and can be exploited to achieve directional translational motion.

As depicted in Fig. 1b, we proposed that in the presence of some electrophilic π-system, X = Y, an aryl–metal species **Int-I** centred at position A will undergo migratory insertion into the X = Y bond preferentially over the A → B [1,2] migration, generating X–Y–metal species **Int-II**. In **Int-II**, the metal centre occupies a position [1,4] in relation to site B and can undergo migration back to the polyaromatic track via

a [1,4] migration, to give **Int-III**. In going from **Int-I** to **Int-III**, the metal centre undergoes a formal [1,2] migration, through sequential migratory insertion–[1,4] migration via a kinetically accessible 5-membered metallocycle. In addition, while the direct [1,2] migration from site A to site B without the incorporation of X = Y would lack directionality, formal [1,2] migration via **Int-II** breaks the symmetry of the system and introduces the possibility of ratcheting the system via the relative energetics and reactivities of intermediates **Int-I**–**III**. From **Int-III**, the metal centre is primed to undergo aryl-to-aryl [1,4] migration across the biaryl bond to site A′, giving **Int-IV**. While such migrations have been described in the literature, directionality associated with the migration (i.e., whether the metal centre favours site B or site A′) has been shown to derive from both electronic and steric differentiation of the two aromatic rings[30–32,38]. Attempts to use electronic bias to control the [1,4] migrations along an extended polyaromatic track are going to achieve limited success as gradients of sufficient electronic differentiation would be required across multiple biaryl units. We postulated, however, that the incorporation of X = Y not only breaks the symmetry of the formal [1,2] migration but also that of the [1,4] migration. The carbon–metal bonds in **Int-III** and **Int-IV** exist in

markedly different steric environments with [1,4] migration to **Int-IV** releasing steric strain associated with the *ortho* X–Y group in **Int-III**. **Int-IV** can then undergo insertion into another equivalent of X = Y and the sequence of migratory insertion–[1,4] migration–[1,4] migration repeats to move the metal centre another aromatic unit along the polyaromatic chain. It is important to note here that alternative pathways are accessible to many of these intermediates. For example, insertion into X = Y may be a competing pathway for **Int-III** and all organo-metal species may undergo competitive demetallation processes (*vide infra*). However, the successful realisation of the design outlined in Fig. 1b would represent an approach to directional translational motion at the molecular level where carbon-to-carbon metal migrations are harnessed to achieve dynamic association of a moving component to a polyaromatic track.

In this work, the moving component—the metal centre—is transported along the length of the polyaromatic track with complete directionality, which arises from the information ratchet mechanism: each molecule of X = Y is introduced in the position preceding the metal centre, which influences the relative rates of the subsequent steps and drives the metal centre in one direction, away from the previously introduced X–Y unit.

## Results

Drawing on an iterative C–H alkylation process reported by Miura and co-workers (Fig. 1c)[35,36], we identified norbornene, with its highly strained C = C bond, in combination with rhodium (Rh) as the basis for the proposed system (i.e., electrophilic π-system, X = Y, and metal, M, respectively). As depicted in Fig. 1c, in Miura's system, iterative incorporation of norbornene around a single aromatic ring arises via sequential cycles of migratory insertion of an aryl-rhodium species into norbornene followed by $sp^3(C)$-to-$sp^2(C)$ [1,4] migration of the rhodium centre back to the *ortho* $C(sp^2)$ position, providing convincing precedent for the first two steps of the three-step cycle of migratory insertion–[1,4] migration–[1,4] migration that underpins the proposed translational motion in Fig. 1b. The final step of this cycle, the [1,4] aryl-to-aryl migration, has only been reported in a stoichiometric system for rhodium[38], although the corresponding catalytic [1,4] aryl-to-aryl palladium migration is well described[29,30], and rhodium has been reported to readily undergo catalytic [1,5] aryl-to-aryl migration[39].

In the system outlined in Fig. 1b, the directional transport of the metal centre along the length of the polyaromatic track is coincident with the directionally sequential (i.e., site selective) modification of the molecular track along which it moves: the metal centre acts processively to functionalise the track at sites A, A′, A″, … in a directionally sequential fashion. Such controlled processive catalysis has long been a subject of intensive research and elegant examples of processive catalysis acting on linear substrates using mechanical bonds have been reported in the literature[46–51]. Limited examples of directionally sequential modification of small molecules have also been reported[52–54], but these iterative functionalisation processes lack inherent processivity and it is thus hard to see how these systems can be exploited to achieve directional transport of the catalytic component, unless through the use of mechanical bonds.

To explore whether the proposed migratory insertion–[1,4] migration–[1,4] migration sequence could facilitate directional transport of a rhodium centre, we designed small biaryl track **1** (Fig. 2a). Track **1** was functionalised with a pinacolato boronic ester (Bpin) to facilitate the loading of the rhodium centre onto the track via a transmetalation event along with a fluorine substituent in the *para* position of the second ring, which we envisaged would limit the incorporation of adjacent norbornane groups on the second aromatic ring and additionally provide a handle for analysis of the reaction by $^{19}$F NMR spectroscopy. Using Miura's conditions as a starting point[35], a brief optimisation provided trisubstituted track **tri-1** in 91% yield by $^1$H

NMR spectroscopy (di: tri substitution = 1.0: >20; see Supplementary Information Table S1 for optimisation details). Isolation of **tri-1** in 84% yield provided material for further analysis and allowed us to confirm the substitution pattern depicted in Fig. 2a, with the norbornane groups distributed across the two aromatic rings at sites 1, 2, and 6, confirming catalysis along the length of the track. The prochirality of the norbornene leads to the generation of **tri-1** as two enantiomeric sets of a 1:1:1:1 mixture of four diastereoisomers, as reflected in the $^{19}$F NMR spectrum (Fig. 2b).

Having established the incorporation of multiple norbornane groups, we sought to confirm 1) the directionally sequential nature of their incorporation, i.e., that the norbornane groups are incorporated at site 1 then site 2 then site 6, and 2) the processivity of the catalysis, i.e., that the Rh centre remains associated with the track throughout the iterative modification process. Confirmation of the processive directionally sequential incorporation of norbornane units along the track represents direct evidence of the directional transport of the Rh centre along the track. To confirm postulates 1 and 2 above, a series of mechanistic experiments were undertaken, as detailed in Fig. 2c–f.

We propose that after the incorporation of the third norbornane group, protonolysis occurs from aryl–Rh species **Int-I-tri-1**, which was borne out when the reaction was conducted in the presence of 25 equivalents of $D_2O$ and D-7-**tri-1** was isolated from the reaction (71% D; Fig. 2c). We reasoned that isolation of lower order substitution products (mono- and di-substituted **di-1**) would allow us to confirm the order in which the norbornane groups are introduced and proposed that these lower order substitution products could be accessed by promoting protonolysis from earlier intermediates through higher concentrations of $H_2O$. Indeed, when the reaction was conducted with increasing concentration of $H_2O$ (14 equiv., 28 equiv., 56 equiv., Fig. 2d) the proportion of **di-1** increased, although interestingly mono-substitution remained only a minor component of the reaction mixture even at high $H_2O$ concentrations. Crucially, we saw no evidence of substitution products that could have arisen from a non-directionally sequential sequence, e.g., di-substitution products **(1,6)-di-1** or **(2,6)-di-1**, which provides strong evidence to suggest that the norbornane groups are introduced in a directionally sequential fashion.

We then sought to confirm that the catalysis was processive, that is, that the rhodium centre remains associated with the track throughout the iterative functionalisation. We subjected di-substituted track **di-1** to the optimised reaction conditions and confirmed that no subsequent functionalisation occurs (Fig. 2e): incorporation of norbornane is contingent on an initial transmetalation loading event and the rhodium centre remaining associated with the track. We also subjected bis-*ortho*-deutero biaryl $D_2$-**1** to the reaction conditions and observed almost complete transfer of deuteration to $D_2$-**tri-1** (97% to 94% D), indicative of a migration event which occurs with complete processivity (Fig. 2f).

We believe that these experiments confirm that 1) the norbornane groups are incorporated in a directionally sequential fashion, and 2) that the rhodium centre remains associated with the track through covalent σ-bonding throughout the catalysis before a final protoderhodation event releases the active catalyst from the track. These two features establish that the rhodium centre moves directionally along the polyaromatic track, confirming autonomous directional molecular-level translational motion in a fully synthetic system in the absence of mechanical bonds.

We then sought to establish the generality of the controlled translational motion and looked at the composition of the track. While literature reports suggest that [1,n] metal migrations can be highly sensitive to the electronic nature of the migration outset and terminus[30,39], our design exploits the incorporation of norbornene to ratchet the translational motion, specifically the [1,4]-$sp^2(C)$-to-$sp^2(C)$ migration, overcoming any bias that derives from the relative electronic properties of the two aromatic rings, $Ar_1$ and $Ar_2$ (Fig. 3a). We

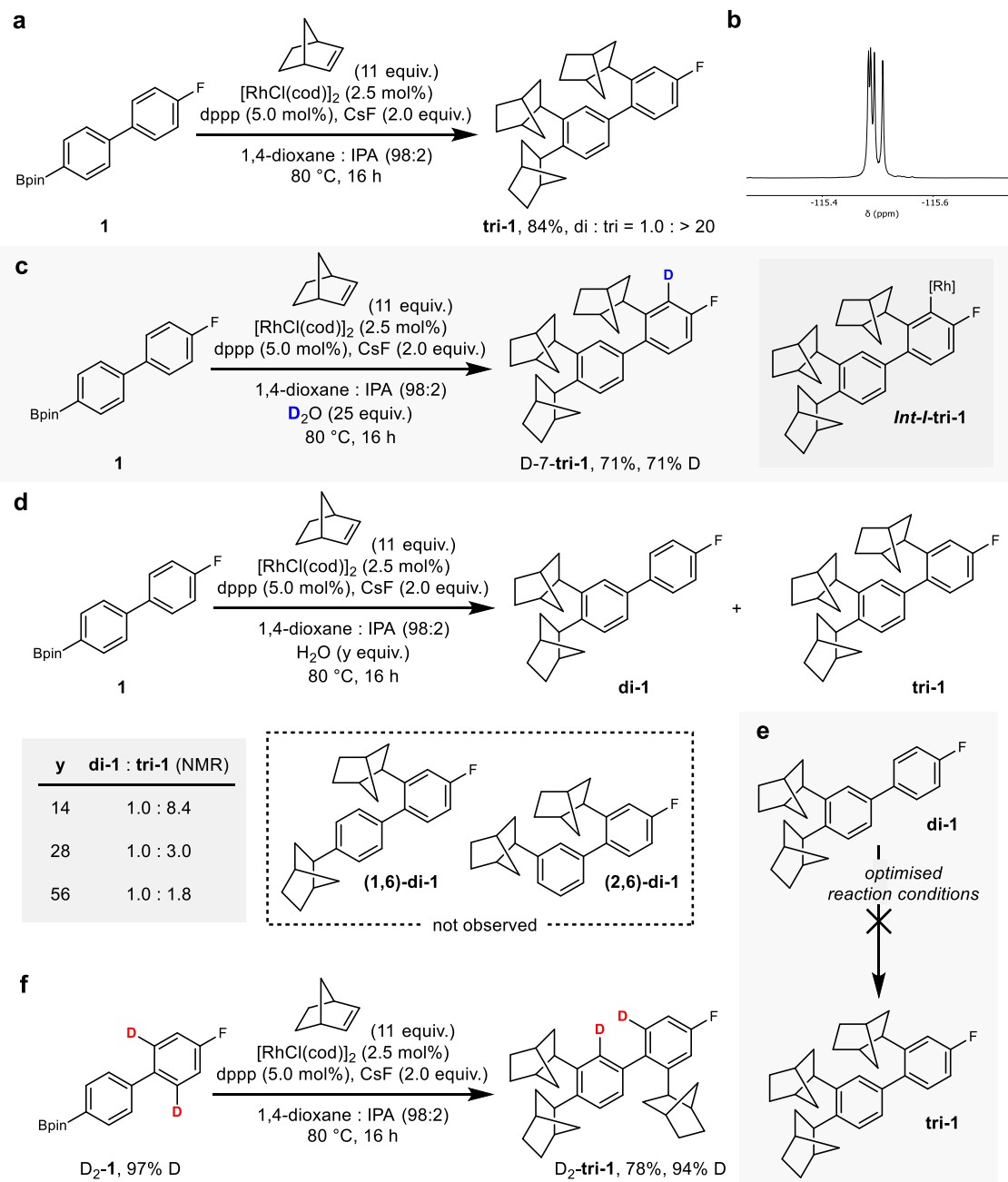

**Fig. 2 | Directionally sequential processive catalysis for controlled directional motion. a** Optimised conditions for the directionally sequential iterative C–H alkylation of **1**: **1** (0.20 mmol), norbornene (11 equiv.), [RhCl(cod)]$_2$ (2.5 mol%), dppp (5.0 mol%), CsF (2.0 equiv.), 1,4-dioxane:IPA (98:2), 80 °C, 16 h; 84% isolated yield of **tri-1**, ratio of substitution products **di-1**: **tri-1** determined by $^1$H NMR spectroscopy of crude reaction. **b** $^{19}$F NMR spectrum of isolated **tri-1**. **c** **1** subjected to optimised reaction conditions in the presence of D$_2$O (25 equiv.); D incorporation determined by $^1$H NMR spectroscopy; blue used to illustrate incorporated D atom. **d** Optimised reaction conditions with increasing equivalents of H$_2$O (y = 14, 28, 56); ratio of substitution products **di-1**: **tri-1** determined by $^1$H NMR spectroscopy of crude reactions. **e** Di-substitution product **di-1** subjected to optimised reaction conditions. **f** Bis-deutero-biaryl **D$_2$-1** subjected to optimised reaction conditions; D incorporation determined by $^1$H NMR spectroscopy; red used to illustrate D labels. Note that all incorporated norbornane units are depicted arbitrarily as (R,R,S).

selected a small library of 7- and 8-substituted biaryl boronic esters (Fig. 3a; **1**–**5**), whose Hammett σ values lay in the range -0.27– + 0.43[55], and subjected these biaryl tracks to the rhodium catalysis. Indeed, the degree of norbornene incorporation, measured by the ratio of di- to tri-substitution products, displays little correlation with the Hammet σ values, confirming that the incorporation of the hydrocarbon is sufficient to overcome any inherent electronic bias in the [1,4]-sp$^2$(C)-to-sp$^2$(C) migration. Interestingly, the degree of norbornene incorporation does appear to be influenced by the steric size of the substituent

on the second aromatic ring, Ar$_2$, as determined by the corresponding A values[56], suggesting that the ratio of di- to tri-substitution products is highly sensitive to steric factors. We also studied a small series of *ortho*-methyl substituted tracks (**6**–**8**). A methyl group in the *ortho* position of the outset ring Ar$_1$ (position 3, **6**; Fig. 3a, Entry 6) has little impact on the migration, nor when the methyl group is moved to the *ortho* position of the terminating ring (position 6, **7**; Fig. 3a, Entry 7). However, when there is a methyl group in the *ortho* position on both the outset and terminating ring (positions 3 and 6, **8**; Fig. 3a, Entry 8), we

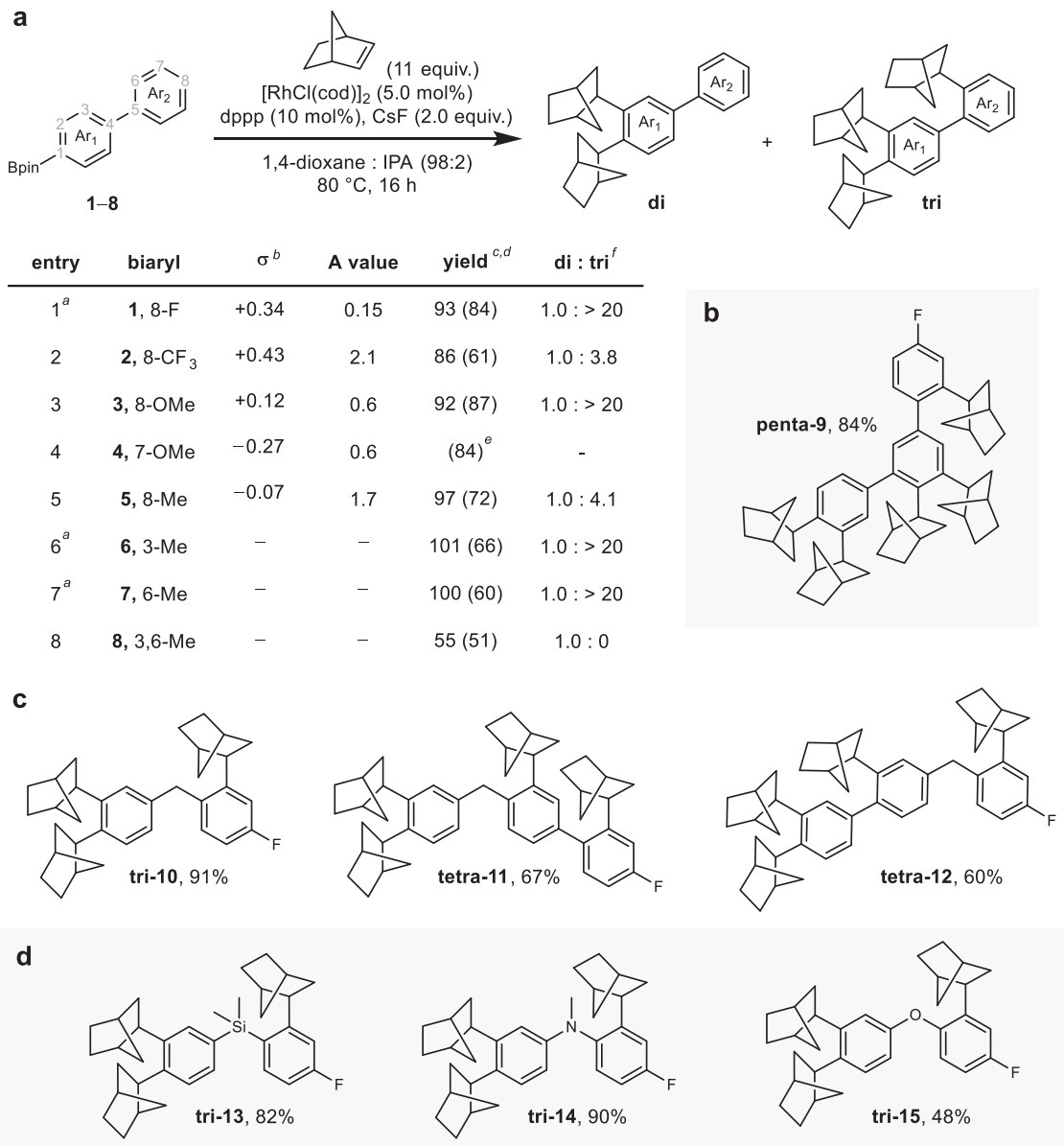

**Fig. 3 | Investigation of track composition for controlled directional motion of Rh centre. a** Library of electronically differentiated biaryl tracks as substrates under optimised reaction conditions: biaryl **1–8** (0.20 mmol), norbornene (11 equiv.), [RhCl(cod)]$_2$ (5.0 mol%), dppp (10 mol%), CsF (2.0 equiv.), 1,4-dioxane:IPA (98:2), 80 °C, 16 h; [a] [RhCl(cod)]$_2$ (2.5 mol%), dppp (5.0 mol%); [b] Hammett σ values are given relative to position 6, i.e., Entries 1–3, 5: σ$_m$, Entry 4: σ$_p$; [c] combined $^1$H NMR spectroscopy yield of di- and tri-substituted products using 1,3,5-trimethoxybenzene as an internal standard; [d] isolated yield of tri-substituted product in parentheses; [e] isolated yield of tetra-substituted product; [f] ratio of substitution products **di**: **tri** determined by $^1$H NMR spectroscopy of crude reactions. **b** Directly linked polyphenylene track with bent conformation, **9**, under optimised reaction conditions with [RhCl(cod)]$_2$ (5.0 mol%), dppp (10 mol%) giving **penta-9**; isolated yield. **c** Methylene-linked biaryl tracks (**10–12**) as substrates under optimised reaction conditions with [RhCl(cod)]$_2$ (2.5 mol%), dppp (5.0 mol%) giving poly-substituted tracks **tri-10, tetra-11, tetra-12**; isolated yields. **d** Me$_2$Si– (**13**), MeN– (**14**), and O– (**15**) linked biaryl tracks as substrates under optimised reaction conditions with [RhCl(cod)]$_2$ (2.5 mol%), dppp (5.0 mol%) giving tri-substituted tracks **tri-13, tri-14, tri-15**; isolated yields. Note that all incorporated norbornane units are depicted arbitrarily as (*R,R,S*).

observe exclusive di-substitution and no evidence of tri-substitution, which we attribute to an inaccessible migration transition state arising due to the sterics associated with bringing the two *ortho*-methyl groups into plane in the migration transition state.

Extension of the polyaromatic track at the 7 position introduced a bent conformation to the track, which was well tolerated, providing the corresponding penta-substituted track **penta-9** in excellent isolated yield (Fig. 3b; see Supplementary Information Section 3.6 for details).

In the design outlined in Fig. 1b, [1,4]-sp$^2$(C)-to-sp$^2$(C) migrations are exploited to achieve directional transport of the metal centre along

a track comprising directly linked aromatic rings. We postulated that incorporating a [1,5]-sp$^2$(C)-to-sp$^2$(C) migration[39], in place of the [1,4]-sp$^2$(C)-to-sp$^2$(C) migration, would allow us to achieve directional transport along more structurally diverse polyaromatic tracks, specifically those incorporating methylene linkages. To our delight, subjecting methylene linked track **10** to the optimised reaction conditions provided the tri-substituted track **tri-10** in 91% yield (Fig. 3c; see Supplementary Information Section 3.7 for details). Crucially, incorporation of both direct and methylene linked aromatic rings into the track is well tolerated leading to the tetrasubstituted tracks **tetra-11** and **tetra-12** in good yield, independent of the order of those linkages

**Fig. 4 | Controlled directional motion of Rh centre over increasing track length.** Controlled translational motion over longer length scales; polyaniline tracks **16** and **17** as substrates under the optimised reaction conditions with [RhCl(cod)]₂ (10 mol %), dppp (20 mol%), norbornene (22 equiv.) giving hexa- and octa-substituted tracks as the major substitution products (**hexa-16** and **octa-17**), respectively; for **16**, isolated combined yield of **tetra-16** and **hexa-16**; for **17**, **octa-17** identified as major component of reaction by MALDI HRMS (see Supplementary Information Section 3.10 for details); shades of green used to illustrate sequential incorporation of norbornene. Note that all incorporated norbornane units are depicted arbitrarily as (R, R, S).

(Fig. 3c). Following the success of methylene linked track **10**, we explored further increases to the structural diversity of the tracks through alternative linking motifs. SiMe₂- and MeN-linked tracks **13** and **14** underwent efficient tri-substitution to give **tri-13** and **tri-14**, respectively, in excellent yield (Fig. 3d), while oxygen-linked track **15** showed a poorer di- to tri-substitution ratio (di: tri substitution = 1.0: 1.4), which we attribute to protoderhodation from the *ortho* position of the first ring becoming competitive with migration across the flexible ether linkage.

Exploiting the highly effective catalysis associated with the MeN-linked track **14** (90%, di: tri substitution = 1.0: >20, Fig. 3d), we then explored tracks of increasing length. To our delight, when polyaniline track **16** is subjected to the reaction conditions in the presence of 22 equivalents of norbornene, hexa-substituted track **hexa-16** is generated as the major component of the reaction with an 80% combined yield of the tetra- (**tetra-16**) and hexa-substituted (**hexa-16**) tracks (tetra: hexa substitution = 1.0: 6.7, Fig. 4). Extending the track by two further aniline units provides penta-aryl track **17**, which when subjected to the reaction conditions (22 equivalents of norbornene) gave octa-substituted track **octa-17** as the major component (~55%) of the reaction, as confirmed by MALDI HRMS (see Supplementary Information Section 3.10 for details of the analysis).

In summary, we have identified carbon-to-carbon rhodium migrations as a strategy for mimicking the dynamic association that underpins controlled translational molecular level motion in biological systems. Directionality is imparted on the translational motion through the repetitive incorporation of a simple hydrocarbon into the polyaromatic track. The reported system represents an example of autonomous directional translational motion at the molecular level in a fully synthetic system that does not rely on mechanical bonds. In addition, it realises a rare example of long elusive directionally sequential processive catalysis[50], and the only example for which processivity is not achieved through mechanical bonds. The system holds great promise in addressing some of the limitations associated with translational motion mediated by mechanical bonds, where track branching is not tolerated, and motion is thus limited to one dimension. The prevalence of metal migrations, and the processivity inherent to their mechanisms, alongside the breadth of possible chemical diversity associated with the system X = Y, suggests that the approach outlined here has the potential to become a general strategy for controlling motion at the molecular level.

## Data availability
The data that support the findings of this study are available within the paper and its Supplementary Information.

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

## Acknowledgements

We thank the Royal Society (RS; University Research Fellowship to B.S.L.C.; URF/R1/180592), the Engineering and Physical Sciences Research Council (EPSRC; studentships to M.N.C. and E.L.H. through the Bristol Centre for Doctoral Training in Technology-Enhanced Chemical Synthesis; EP/S024107/1), the Hilmar Johannes Backer Foundation (scholarship to C.L.F.v.B.), and Adam Nobel and Tim Gallagher for valuable discussions and supervisory guidance of M.N.C. during B.S.L.C.'s maternity leave.

## Author contributions

B.S.L.C. conceived the project. E.L.H., M.N.C., C.L.F.v.B. and P.J.G. designed and carried out the experiments. B.S.L.C. directed the research. All authors contributed to the analysis of the results and the writing of the manuscript.

## Competing interests

The authors declare no competing interests.
