## [Transparent Peer Review file · Nature Communications]

[1,*n*]-Metal migrations for directional translational motion at the molecular level.

Corresponding Author: Dr Beatrice Collins

Version 0:

Reviewer comments:

Reviewer #1

(Remarks to the Author)

In their manuscript “[1,*n*]-Metal migrations for directional translational motion at the molecular level”, Hollis et al. report that the combination of formal [1,2] and [1,4]/[1,5] migrations of a Rh catalyst can be used to induce a sequential and unidirectional motion of the metal center on oligoaromatic tracks. The directional process is driven by an exergonic C-H alkylation with inherently strained norbornene. The resulting norbornane substituents not only inhibit the reverse reaction but also act as “breadcrumb trail” to trace the motion of the Rh metal center. After a thorough optimization of reaction conditions, the authors provide several experiments to get mechanistical insights into their system using a simple biphenyl track and then increase the complexity by elongation the track with further aromatic units. The experiments in this study are thorough, methodologically sound and well-executed.

In my opinion, the concept is highly creative and introduces a completely new approach to achieve directed motion at the molecular scale, thus having a high potential to find application in complex molecular machines. I consider the insights from this study to be highly relevant to various fields of chemistry, including molecular machinery and catalysis.

I have only a few minor comments, mostly to improve the intelligibility of the manuscript:

1) From Fig. 1 alone, I think it is challenging to grasp the concept of the study. In my opinion, Fig. 1a, b and d are not entirely necessary, while Fig. 1c should be presented in more detail to guide the reader through the mechanism of the directional motion. I would suggest that the authors show the full reaction mechanism step-by-step (including transmetalation, sequential migration and protolysis and maybe even 5/6-membered metallacycles) on a representative, borylated bi- or triphenyl track. This would help readers from different research areas in chemistry to better understand the complex concept.

2) Since the study is highly relevant for molecular machinists, I would appreciate a more in-depth discussion of the ratcheting mechanism present in this system.

3) Penta-9 is the main product after alkylation (figure 3b). I was wondering if tri-9 (resulting from two consecutive [1,4] migrations) was also observed?

4) A potential downside of the system is that the track can only be used once, as it is alkylated during Rh migration. Is there a way to reset the track?

In conclusion, I think that the study is intriguing, creative and exceptionally well done, making it perfectly suited for the broad scientific readership of this journal. I therefore highly recommend the manuscript for publication in Nature Communications.

Reviewer #2

(Remarks to the Author)

In this manuscript, Collins and colleagues presented a new platform for dynamic correlation based on carbon-carbon metal migration. The successful repetitive doping of simple olefins into aromatic orbitals represents an exciting case for converting previously elusive processes into directed sequential catalysis. A great deal of work has also been done in demonstrating

reaction adaptability and elaborating the catalytic process. But the reaction strategy was reported by Miura and co-workers (J. Am. Chem. Soc. 2000, 122, 10464–10465, DOI: 10.1021/ja002601w). The catalytic process described in this manuscript is very similar with Miura's method. Even the catalyst, ligand and base are also the same with the reported reference. This manuscript only extended the reaction scope to substituted biaryl boronic esters and investigated the mechanism for this reaction.

This innovation of this work is not good enough, I cannot support publication in a high-impact journal like Nature Communications. I therefore suggest rejection and publication elsewhere, such as Org. Lett..

There are some errors of detail in the manuscript and some issues need to be addressed.

1. For demonstrating the importance of this reaction, the authors would investigate the substrate scope for norbornene. The rigid structure is very special, whether the norbornene derivatives are suitable for this reaction.
2. The H-source of the product from C=C to C-C for norbornene should be investigated. Whether it is from solvent isopropanol or water?
3. Some metal migrations references and reviews should be cited (Review: Angew Chem. Int. Ed. DOI:10.1002/anie.201913382; Chin. J. Chem. 2019, 37, 929-945; Org. Chem. Front. 2020, 7, 3530-3556), for Co: J. Am. Chem. Soc. 2024, 146, 26223–26232; Org. Lett. 2014, 16, 3392–3395; for Pd: J. Am. Chem. Soc. 2021, 143, 1641–1650; J. Am. Chem. Soc. 2024, 146, 18811–18816; for Ni: JACS, DOI:10.1021/jacs.0c08810; DOI: 10.1002/anie.202304447; DOI: 10.1002/anie.202304713.
4. In the abstract section, the authors present carbon-carbon metal migration with the help of motor protein movement is a creative angle. However, the abstract is too long, and the introduction of motor proteins is less relevant to the main text.
5. For the sake of completeness of the manuscript, it is recommended that the authors add the screening process for the optimal conditions.
6. On the second page, the term “chemical fuel” appears on line 20, and it is recommended that a specific description be added to allow the reader to visualize its meaning.
7. Some of the text in the manuscript does not need to be italicized, such as the last line of page 3, “directional”, and page 9, line 6, “then”.
8. The “note that” in line 16 on page 4 should be stated as “noting that”.
9. Page 6, line 24, Scheme 2a should read Figure 2a.
10. On page 9, line 4, at the end of the line, there is an additional “1”.
11. Page 11, line 12, for “ortho methyl” it should read “ortho-methyl”
12. Page 13, line 17, the yield of the product tri-15 is inconsistent with the description in Fig. 3.
13. In Figure 4, it is recommended that the corresponding yields of the reaction product octa-17 be indicated in the graph.
14. Refs. 13, 44, journal titles should be abbreviated.

The following are the revisions in the supporting information

1. S3: In the second line of the second paragraph, a space is missing between “10” and “g”.
2. S3: In the third line of the third paragraph, the “C NMR” should be stated as “¹³C NMR”.
3. S6: In the third line of the second paragraph, the word size of –62.37 is too small, which is different from other numbers.
4. S19: In the penultimate line of the last paragraph, “⁹F NMR” should be written as “¹⁹F NMR”.
5. S71, Rf (hexane) information is missing.
6. S67, the “¹H NMR” on page S67 should be in bold font.
7. The baselines of the NMR spectra of some compounds are not straight lines, e.g., di-1, tri-15. Modifications are recommended to avoid these defects.
8. The NMR spectra of some of the compounds were not processed by correct integration. For example, D2-S1a, D2-S1b, D2-S1.
9. The NMR spectrum of compound di-1 lacks the structure of the corresponding compound

Reviewer #3

(Remarks to the Author)

This contribution from Collins and coworkers has identified carbon-to-carbon rhodium migrations as a new approach to mimicking the dynamic association that underpins controlled translational molecular level motion in biological systems. The reported system represents the first example of autonomous directional translational motion at the molecular level in a fully synthetic system that does not rely on mechanical bonds. The system holds great promise in addressing some of the limitations associated with translational motion mediated by mechanical bonds, where track branching is not tolerated, and motion is thus limited to one dimension. A reasonable substrate scope is demonstrated. The quality of the SI is good.

Overall, I support publication of this manuscript in Nat. Commun. after the minor points below have been addressed.

- 1) Currently, the acceptor is limited to norbornene, could any other acceptor (for example, alkynes, cod, NBD) be suitable?
- 2) The authors are suggested to use deuterated d1-IPA to see if the deuterated product could be obtained.
- 3) Does the author test the function of this kind of multi-substituted polyaniline track? It seems that this is a suitable method to modify polyarenes.
- 4) For the simple Ph-Bpin, could tri- or tetra- products be obtained.

Version 1:

Reviewer comments:

Reviewer #1

(Remarks to the Author)

The authors have addressed all my comments and thoroughly answered my, mainly curiosity-driven, questions to my full satisfaction. The concept of the manuscript is now explained in much greater detail and is easier to follow. In my opinion, the study is meticulously conducted and the combination of [1,2] and [1,4] migrations to achieve an unprecedented directed translational motion along a chemical track via a “bridge-burning walking mechanism” is highly innovative and simply brilliant. I am confident that this work will be extremely well received by the molecular machines community. Given the importance of this study, I strongly recommend its publication in a high-impact, general scientific journal such as Nature Communications.

Reviewer #2

(Remarks to the Author)

I am pleased to note that the authors have comprehensively addressed all the reviewers' comments. Additional experiments have been conducted to further substantiate the novelty of this manuscript. Having thoroughly reviewed the revised manuscript and the authors' responses, I now have a complete understanding of the work. All the issues I previously raised have been satisfactorily resolved. I find this manuscript to be of significant importance and would strongly support its publication in Nature Communications.

Reviewer #3

(Remarks to the Author)

In this revised manuscript, the authors provide substantial revisions to address the prior concerns in detail. Therefore, I would like to recommend the acceptance of this work in Nat. Commun. without revision.

Detailed responses to the reviewers' comments

We have addressed the reviewers' comments in turn (#1 to #3) and have numbered the concerns we have identified.

Reviewer #1:

We are delighted that Reviewer #1 considers the study to be *intriguing, creative and exceptionally well done* and that is *perfectly suited for the broad scientific readership of ... Nature Communications*. Below we address their additional comments.

1. From Fig. 1 alone, I think it is challenging to grasp the concept of the study. In my opinion, Fig. 1a, b and d are not entirely necessary, while Fig. 1c should be presented in more detail to guide the reader through the mechanism of the directional motion. I would suggest that the authors show the full reaction mechanism step-by-step (including transmetalation, sequential migration and protolysis and maybe even 5/6-membered metallacycles) on a representative, borylated bi- or triphenyl track. This would help readers from different research areas in chemistry to better understand the complex concept.

We thank Reviewer #1 for this comment and for flagging that the complex concept is at present challenging to grasp. We would like to balance Reviewer #1's suggestions of revisions for Figure 1 with the comments from Reviewer #2 regarding the relationship of the reported system with the report from Miura and co-workers in 2000 [*J. Am. Chem. Soc.* **122**, 10464–10465 (2000)] (point 5), and the significance of the broader concept of the system to the non-specialist reader (Reviewer #2, point 9).

In light of these competing demands on a single figure and the limited space available to address them all, we have introduced a 'cartoon' style schematic to accompany Figure 1c (now Figure 1b); we believe that this additional schematic helps to both guide the reader through the detailed catalytic mechanism (addressing Reviewer #1's point 1) and reinforces the broader concept of translational motion, which we believe is valuable in light of Reviewer #2's point regarding the abstract (point 9). In line with Reviewer #1's suggestion, we have removed Figure 1b, although we have incorporated the concepts originally illustrated there into Figure 1a, as we believe this framing is essential to highlight the significance and challenges of the work.

We have also retained Figure 1d (now Figure 1c) to highlight the importance of Miura's report to key steps of the catalysis (Reviewer #2, point 5).

In addition, we have included a new scheme in the Supplementary Information (Scheme S1, page S94). This new figure provides a detailed discussion of the mechanism of the directional motion, using the standard biaryl track **1** as a representative track, as suggested explicitly by Reviewer #1. The new figure includes the transmetalation, sequential migration, and protonolysis steps along with the key 5-membered metallocycles, which we believe will help readers from different research areas in chemistry to better understand the complex concept.

2. Since the study is highly relevant for molecular machinists, I would appreciate a more in-depth discussion of the ratcheting mechanism present in this system.

We agree with Reviewer #1 that a more in-depth discussion of the ratcheting mechanism present in the system would be valuable. Prompted by this comment we have introduced additional discussion at the end of the section on the generalised system (e.g., page 4, lines 19–27 and page 5, lines 1–23). The final two sentences of this section now read:

“However, the successful realisation of the design outlined in Figure 1b would represent a new approach to directional translational motion at the molecular level where, for the first time, carbon-to-carbon metal migrations are harnessed to achieve dynamic association of a moving component to a polyaromatic track. The moving component—the metal centre—is transported along the length of the polyaromatic track with complete directionality, which arises from the information ratchet mechanism: each molecule of X=Y is introduced in the position preceding the metal centre, which influences the relative rates of the subsequent steps and drives the metal centre in one direction, away from the previously introduced X–Y unit.”

3. Penta-9 is the main product after alkylation (figure 3b). I was wondering if tri-9 (resulting from two consecutive [1,4] migrations) was also observed?

We thank Reviewer #1 for this insightful question. We isolate only two components from the reaction mixture: **penta-9** and the protodeborylated starting material (**PDB-9**) in 84% and 8% yields, respectively. We have now included the additional data for the isolation of **PDB-9** in the Supplementary Information (S77). While we cannot definitively rule out the formation of

small quantities of **tri-9** (the signals corresponding to such a species would likely overlap with the signals for **penta-9** in the ^1H NMR spectrum of the crude reaction mixture), we note that due to its likely similar polarity to that of **penta-9** and **PDB-9** we would expect to isolate **tri-9** if it were present in any significant quantity. We attribute the preference for **penta-9** over **tri-9** to the relative steric congestion at the two possible migration termini on the central ring and the resultant relative barriers to the C–H activation events for the competing C–H bonds. Controlling whether directional motion proceeds along the inside or outside of the track is highly attractive and future studies will explore structural derivatives of **9** aimed at promoting the ‘shortcut’ pathway, although this is beyond the scope of the current study.

4. A potential downside of the system is that the track can only be used once, as it is alkylated during Rh migration. Is there a way to reset the track?

We thank Reviewer #1 for this perceptive comment. Indeed, the major limitation of this system is that it is characterised by a ‘bridge burning’ mechanism, whereby directional motion of the Rh centre is coupled to deactivation of the track for further motion. While beyond the scope of the current system, future studies will focus on the identification of alternative iterative C–H functionalisation processes using different X=Y units, which would allow either spontaneous or controlled subsequent defunctionalisation and thus the ‘re-setting’ of the track. The evolution of bridge-burning mechanisms into fully sustainable systems for other nanotechnologies, for example, DNA nanowalkers, has been well documented in the literature [Wang, Z., Hou, R., Lo, I. Y. *Nanoscale* **11**, 9240–9263 (2019)].

Reviewer #2:

5. But the reaction strategy was reported by Miura and co-workers (J. Am. Chem. Soc. 2000, 122, 10464–10465, DOI: 10.1021/ja002601w). The catalytic process described in this manuscript is very similar with Miura’s method. Even the catalyst, ligand and base are also the same with the reported reference. This manuscript only extended the reaction scope to substituted biaryl boronic esters and investigated the mechanism for this reaction.

This innovation of this work is not good enough, I cannot support publication in a high-impact journal like Nature Communications. I therefore suggest rejection and publication

elsewhere, such as Org. Lett..

We are pleased that Reviewer #2 comments that the *successful repetitive doping of simple olefins into aromatic orbitals represents an exciting case for converting previously elusive processes into directed sequential catalysis* and that *a great deal of work has also been done in demonstrating reaction adaptability and elaborating the catalytic process*. Some steps of the catalytic process that underpins the unprecedented directional translational motion of the Rh centre over the linear tracks are indeed similar to the report from Miura and co-workers in 2000 [*J. Am. Chem. Soc.* **122**, 10464–10465 (2000)]. We make this explicitly clear in the manuscript, referencing Miura by name on four occasions and dedicating a section of the first figure (Figure 1d, now Figure 1c) to that methodology. We acknowledge Referee #2's opinion on this point but would like to raise a couple of counterarguments.

The novelty of the reported work lies not with the catalytic methodology but with the fact that the work introduces a fundamentally new approach to controlled molecular-level translational motion. Efforts to develop fully synthetic systems that exhibit controlled—*directional*—motion of a moving molecular component relative to a linear molecular track have been the subject of considerable research attention over the last thirty years, but almost all contributions to the field have been based on rotaxane or pseudo-rotaxane architectures where the key property of association between the moving and static molecular components is achieved with mechanical bonds. The handful of systems that have been developed that are not based on rotaxane-type architectures lack the crucial behaviours that underpin the promise of such systems, namely, directionality and autonomous behaviour. This work represents the first example of autonomous directional translational motion at the molecular level in a fully synthetic system that does not rely on mechanical bonds and, as a result, the fundamental novelty that characterises this work within a field that has retained scientific prominence over multiple decades makes it a significant scientific advance of immediate importance and, we believe, suitable for publication in *Nature Communications*. We also note that Reviewer #1 comments that *“the concept is highly creative and introduces a completely new approach to achieve directed motion at the molecular scale, thus having a high potential to find application in complex molecular machines”*. Reviewer #3 also acknowledges this fundamental novelty where they comment that the *“reported system represents the first example of autonomous directional translational motion at the molecular level in a fully synthetic system that does not rely on mechanical bonds”*.

We would like to raise a number of further points in response to Reviewer #2's comments. Firstly, while the incorporation of norbornene under rhodium catalysis draws on Miura's report from 2000, the catalytic [1,4] aryl-to-aryl rhodium migration across the directly linked biaryls, which is also crucial to achieving the extended translational motion of the rhodium centre, has not been previously reported (see page 6, lines 7–10 of the manuscript for discussion). We would also like to correct Review #2's statement that the work "*only extended the reaction scope to substituted biaryl boronic esters*". Considerable work is also presented for the catalysis on extended polyaromatic tracks (e.g., tracks **9**, **11**, **12**, **16** and **17**).

Furthermore, as outlined in the introductory discussion of the manuscript, we believe that this report has the potential to become a general strategy for achieving autonomous directional translational motion at the molecular level, where alternative electrophilic π -systems ($X=Y$) could be exploited to achieve the key dynamic association of the metal centre to the polyaromatic track that is fundamental to achieving directional translational motion at the molecular level. In this report we have shown norbornene to be an effective candidate for $X=Y$, but we do not believe the system is limited to norbornene incorporation and the realisation of the general strategy with alternative $X=Y$ and metal combinations is the subject of continued study.

Finally, it is important to note that within the field of molecular machines, the exploitation of chemical processes previously reported in the literature to achieve wholly novel function or behaviour is well precedented, where many systems being reported at the highest level exploit chemical transformations which have direct precedent in the literature. For example, the fuelled formation and cleavage of anhydride intermediates that underpins the seminal autonomous chemically-fuelled single bond rotary motor recently reported by Leigh and co-workers [Borsley, S., Kreidt, E., Leigh, D. A., Roberts, B. M. W. *Nature* **604**, 80–85 (2022)] has extensive precedent within the field of dissipative assembly [e.g., Kariyawasam, L. S., Hartley, C. S. *J. Am. Chem. Soc.* **139**, 11949–11955 (2017)], as highlighted by the authors.

6. For demonstrating the importance of this reaction, the authors would investigate the substrate scope for norbornene. The rigid structure is very special, whether the norbornene derivatives are suitable for this reaction.

Reviewer #2 is correct in their observation that *the rigid structure [of norbornene] is very special*. We can confirm that simple unstrained alkenes, such as cyclohexene, show no reactivity, leading to complete protodeboronation of the starting polyaromatic tracks. This experiment has been included in the optimisation table that we have introduced into the SI (Table S1, S3.4, S59–S60) in response to Reviewer #2's suggestion (point 8 below). While investigations into norbornene derivatives are indeed of interest, they are beyond the scope of the current study. Substituted norbornene derivatives introduce additional stereo- and regio-selectivity considerations which result in increasingly complex mixtures of isomers of the polysubstituted aromatic tracks, which render analysis even more challenging. We note that the major claim of the current study—the first example of autonomous directional translational motion at the molecular level in a fully synthetic system that does not rely on mechanical bonds—would be unchanged by the use of norbornene derivatives.

7. The H-source of the product from C=C to C-C for norbornene should be investigated. Whether it is from solvent isopropanol or water?

When the system is operating optimally, i.e., the Rh centre is undergoing processive directional translation along the polyaromatic track, the source of “H” for the transformation of C=C to C–C is the aromatic C–H bond adjacent to the site of substitution. If the source of “H” for the transformation of C=C to C–C was from isopropanol or water then the rhodium centre would dissociate from the track and directional translational motion would be precluded (as detailed in the discussion of the mechanistic experiments in Figure 2 in the manuscript). Protoderhodation does however represent the key terminating event for the translational motion of the Rh centre (which in turn allows catalytic turnover) and we believe that this protoderhodation arises from adventitious water rather than the IPA co-solvent, as discussed in more detail in response to comment 14 from Reviewer #3 (see below). We believe that these mechanistic details are now clearer following the inclusion of an additional Figure in the Supplementary Information (Scheme S1, page S94), which details the possible mechanistic pathways for standard biaryl track **1** (included in response to Reviewer #1's comment (point 1)).

8. Some metal migrations references and reviews should be cited (Review: *Angew Chem. Int. Ed.* DOI:10.1002/anie.201913382; *Chin. J. Chem.* 2019, 37, 929-945; *Org. Chem. Front.* 2020, 7, 3530-3556), for Co: *J. Am. Chem. Soc.* 2024, 146, 26223–26232; *Org. Lett.* 2014, 16, 3392–

3395; for Pd: J. Am. Chem. Soc. 2021, 143, 1641–1650; J. Am. Chem. Soc. 2024, 146, 18811–18816; for Ni: JACS, DOI:10.1021/jacs.0c08810; DOI: 10.1002/anie.202304447; DOI: 10.1002/anie.202304713.

We have included the additional references and reviews on metal migrations suggested by Reviewer #2 (except the third suggested reference, which was already present) and these additions can be seen in the revised manuscript with tracked changes.

9. In the abstract section, the authors present carbon-carbon metal migration with the help of motor protein movement is a creative angle. However, the abstract is too long, and the introduction of motor proteins is less relevant to the main text.

We thank Reviewer #2 for their comments on the abstract. We believe that the analogy drawn with translational motion in biological systems is crucial in order to highlight the challenges associated with achieving directional translational motion in fully synthetic molecular systems. The property of ‘dynamic association’ which is critical in achieving directional translational motion, and for which we believe this report provides a wholly novel solution in the form of carbon-to-carbon migrations, is illuminated by the comparison to biological systems, where the motion of motor proteins along one dimensional cytoskeletal tracks is characterised by dynamic association through a variety of complex mechanisms. In addition, by highlighting the relationship between translational motion and the functional behaviour displayed by motor proteins, the significance of achieving controlled translational motion in a fully synthetic molecular system and its promise in the development of future nanoscale technologies is established. It is also worth noting that drawing inspiration from biological systems in the design of novel molecular machines is well precedented within the field and this approach has been critiqued by Leigh and co-workers [Zhang, L., Marcos, V., Leigh, D. A. Molecular machines with bio-inspired mechanisms. *Proc. Natl. Acad. Sci.* **115**, 9397–9404 (2018)]. For these reasons we suggest that the current focus of the abstract is appropriate both for the specialist field and for a broader general scientific audience.

10. For the sake of completeness of the manuscript, it is recommended that the authors add the screening process for the optimal conditions.

In response to Reviewer #2's comment we have included a table (Table S1; S3.4, S59–S60) in the Supplementary Information that details selected optimisation studies for the development of the rhodium-catalysed incorporation of norbornene into 2-(4'-fluoro-[1,1'-biphenyl]-4-yl)-4,4,5,5-tetramethyl-1,3,2-dioxaborolane (**1**). Key control experiments are also included in Table S1 along with studies using alternative X=Y systems (see points 6 and 14 from Reviewers #2 and #3, respectively).

11. On the second page, the term “chemical fuel” appears on line 20, and it is recommended that a specific description be added to allow the reader to visualize its meaning.

We thank Reviewer #2 for highlighting our use of the term “chemical fuel” in the manuscript. We have removed this term and the sentence now reads “*In addition to exploiting the dynamic association exhibited by metal migrations, we use the incorporation of a simple hydrocarbon to ratchet the system and achieve the directional translation of a rhodium centre over extended polyaromatic tracks.*” (page 2, lines 19–22). In addition we have removed the term “fuel” in five additional instances, replacing it with the term “hydrocarbon” (all instances are recorded with tracked changes in the revised manuscript and detailed in the list of line edits document). While the term “chemical fuel” has been used extensively in the literature in recent years, there have been recent objections to its widespread usage [Aprahamian, I., Goldup, S. M. *J. Am. Chem. Soc.* **145**, 14169–14183 (2023)] and we are of the opinion that the use of the term is not illuminating in the context of the reported system and thank Reviewer #2 for highlighting its use.

12. Some of the text in the manuscript does not need to be italicized, such as the last line of page 3, “directional”, and page 9, line 6, “then”.

The italicization of the terms “directional” (page 3, line 17) and “then” (page 9, line 10) was undertaken to provide emphasis in the text. However, we are happy to remove this formatting and allow the emphasis to be inferred from the context alone. These revisions are recorded with tracked changes in the revised manuscript and detailed in the list of line edits document

13. We thank Reviewer #2 for the additional detailed comments they have provided on both the manuscript and Supplementary Information. These have been implemented where appropriate and are recorded in the tracked changes on those revised documents; they are also

detailed in the additional document listing the individual line edits undertaken. Revisions requiring further comment are detailed below:

(a) Reviewer #2 comments that “ *“note that” ... should be stated as “noting that”* ”. We are using “note that” in the imperative throughout the manuscript and believe its use in this way to be appropriate, although we are happy to receive further editorial advice on this point.

(b) Reviewer #2 comments that there is an additional “1)” at the end of line 4 on page 9; this is the beginning of a two item list and is followed by “2)” on line 6.

(c) Reviewer #2 notes that the yield of the product **tri-15** on page 13, line 17 is inconsistent with the description in Fig. 3. This is incorrect although we acknowledge that the way this has been reported is not entirely clear and thank Reviewer #2 for bringing this to our attention. The yield given in parentheses on page 13, line 17 (83%) is the combined yield of isolated **di-15** and **tri-15** (35% and 48%, respectively, see S87), while the yield given in Figure 3d refers to the isolated yield of **tri-15** only (48%), in line with the isolated yields reported for the series **tri-13**, **tri-14**, **tri-15**. While the isolated yield of **tri-15** can be calculated from the combined yield of 83% and the ratio of **di-15** : **tri-15** also provided in the parentheses on page 13, line 17, we acknowledge that the discrepancy between the yields reported could be confusing, so have removed the yield 83% from page 13, while leaving the ratio of **di-15** : **tri-15**. The full experimental details are given in the Supplementary Information (Table S16, S87).

(d) Reviewer #2 has recommended that the corresponding yields of the reaction product octa-17 be indicated in the graph in Figure 4. We would like to clarify that the value of “55%” given on line 23, page 14 (“... gave octa-substituted track **octa-17** as the major component (approximately 55%) of the reaction, as confirmed by MALDI HRMS (see Supplementary Information Section 3.8 for details of the analysis)”) corresponds to an approximate value of percentage component by ion yield as desorbed from the original solid sample and does not correspond to an absolute yield. The important claim being made here, supported by the MALDI HRMS data provided in the Supplementary Information (S3.10.2, page S97–S98), is that the major substitution product from the reaction is **octa-17** from which we conclude that the Rh centre has undergone directional translational motion along the length of the pent-aryl track **17** to a significant degree. We believe that this claim is appropriately conveyed in Figure 4 with the label “**octa-17** major component by MALDI HRMS”, alongside the discussion in

the text (page 14, lines 20–24) and the associated section of the Supplementary Information (S3.10.2, page S97–S98).

(e) Reviewer #2 has noted that “The baselines of the NMR spectra of some compounds are not straight lines, e.g., di-1, tri-15. Modifications are recommended to avoid these defects.” The spectra have been corrected with a polynomial fit method (polynomial order 15) to address this issue and this fitting method has been detailed in the relevant section of the General Experimental Details of the Supplementary Information (S4) and alongside all ^{13}C NMR data to which this fitting method has been applied.

Reviewer #3:

We are delighted that Reviewer #3 recognises that the “*reported system represents the first example of autonomous directional translational motion at the molecular level in a fully synthetic system that does not rely on mechanical bonds*” and that they highlight that the “*system holds great promise in addressing some of the limitations associated with translational motion mediated by mechanical bonds*”. Below we address their additional points.

14. Currently, the acceptor is limited to norbornene, could any other acceptor (for example, alkynes, cod, NBD) be suitable?

We thank Reviewer #3 for this point, which is related to point 6 from Reviewer #2 above. As detailed above, simple unstrained alkenes, such as cyclohexene, show no reactivity, leading to complete protodeboronation of the starting polyaromatic tracks. The use of norbornadiene (NBD), as suggested by Reviewer #3, leads to a complete shutdown of catalytic reactivity and return of the starting biaryl boronic ester track; we attribute this to the well-known ligand properties of NBD for the Rh centre. These experiments have now been included in the optimisation table that has been introduced into the Supplementary Information (Table S1, page S59–S60). While investigations into the use of alternative acceptors such as alkynes and COD, as suggested by Reviewer #3, are beyond the scope of this current study, where their use will not fundamentally impact the claims of the present work, future studies into the behaviour of

alternative acceptors are of considerable interest and we thank Reviewer #3 for these suggestions.

15. The authors are suggested to use deuterated d1-IPA to see if the deuterated product could be obtained.

We have undertaken the experiment in which the IPA co-solvent is exchanged for d8-IPA (Reviewer #3 specifically suggests the use of d1-IPA but both reagents have the key protic deuterium atom that we understand Reviewer #3 is suggesting should be interrogated). This experiment, now included in the Supporting Information (S3.8.2, page S91), leads to no incorporation of deuterium. This in contrast to the experiment conducted with D₂O (Figure 2c of the manuscript, S3.8.1, page S89–S90), in which the inclusion of 25 equivalents of D₂O results in high levels of deuterium incorporation (71% D incorporation as determined by ¹H NMR spectroscopy). From these experiments we conclude that the key terminating protoderhodation events arise from adventitious H₂O rather than the IPA co-solvent. We have included a sentence in the Supporting Information (S91; “*The absence of deuterium incorporation observed in the presence of d8-IPA in contrast to the high levels of deuterium incorporation that arise when the reaction is conducted in the presence of 25 equivalents of D₂O (S3.8.1) lead us to conclude that the key terminating protoderhodation events arise from adventitious H₂O rather than the IPA co-solvent.*”) summarising these findings.

16. Does the author test the function of this kind of multi-substituted polyaniline track? It seems that this is a suitable method to modify polyarenes.

We thank Reviewer #3 for the observation that the reported system would be *a suitable method to modify polyarenes*. A comprehensive investigation into the properties of the regularly alkylated polyanilines that are generated in tandem to the directional translational motion of the Rh centre would be of considerable value, particularly as polyanilines are well known to exhibit high electrical conductivity and their extensive applications have been the subject of comprehensive review [Zare, E. N., Makvandi, P., Ashtari, B., Rossi, F., Motahari, A., Perle, G. J. *J. Med. Chem.* **63**, 1–22 (2020)]. We believe that such a study is beyond the scope of the current report, which focuses on the directional motion of the Rh centre, but hope to explore this aspect of the system, in particular in light of alternative X=Y acceptors (see point 14 from Reviewer #3 above) in future studies.

17. For the simple Ph-Bpin, could tri- or tetra- products be obtained.

In agreement with Miura's reported results [*J. Am. Chem. Soc.* **122**, 10464–10465 (2000)], subjecting PhB(OH)₂ to reaction conditions closely analogous to those ultimately used for the polyaromatic Bpin tracks provided a mixture of substitution products, including both tri- and tetra-substituted benzene. As detailed in Table S1 (page S59), we have observed identical results for the B(OH)₂ and Bpin biaryl track **1** and infer that this would also be the case for PhB(OH)₂ and PhBpin.